# BrainRAM: Cross-Modality Retrieval-Augmented Image Reconstruction from Human Brain Activity

### Dian Xie
Hefei University of Technology
Hefei, Anhui, China
xiedian@mail.hfut.edu.cn

### Peiang Zhao
University of Science and Technology of China
Hefei, Anhui, China
pazhao@mail.ustc.edu.cn

### Jiarui Zhang
Hefei University of Technology
Hefei, Anhui, China
2022217188@mail.hfut.edu.cn

### Kangqi Wei
Hefei University of Technology
Hefei, Anhui, China
kangqiwei.work@gmail.com

### Xiaobao Ni
Hefei University of Technology
Hefei, Anhui, China
gsakanama@gmail.com

### Jiong Xia[*]
Hefei University of Technology
Hefei, Anhui, China
xiajiong@mail.hfut.edu.cn

## ABSTRACT

Reconstructing visual stimuli from brain activities is crucial for deciphering the underlying mechanism of the human visual system. While recent studies have achieved notable results by leveraging deep generative models, challenges persist due to the lack of large-scale datasets and the inherent noise from non-invasive measurement methods. In this study, we draw inspiration from the mechanism of human memory and propose BrainRAM[1], a novel two-stage dual-guided framework for visual stimuli reconstruction. BrainRAM incorporates a Retrieval-Augmented Module (RAM) and diffusion prior to enhance the quality of reconstructed images from the brain. Specifically, in stage I, we transform fMRI voxels into the latent space of image and text embeddings via diffusion priors, obtaining preliminary estimates of the visual stimuli's semantics and structure. In stage II, based on previous estimates, we retrieve data from the LAION-2B-en dataset and employ the proposed RAM to refine them, yielding high-quality reconstruction results. Extensive experiments demonstrate that our BrainRAM outperforms current state-of-the-art methods both qualitatively and quantitatively, providing a new perspective for visual stimuli reconstruction.

## CCS CONCEPTS

• **Computing methodologies** → **Visual content-based indexing and retrieval**; **Reconstruction**; **Cognitive science**.

## KEYWORDS

Retrieval-Augmented Generation, Neural Decoding, Brain-Computer Interface

---

[*]Corresponding Author.

[1]Code is available at https://github.com/HQ406/BrainRAM.

---

**ACM Reference Format:**
Dian Xie, Peiang Zhao, Jiarui Zhang, Kangqi Wei, Xiaobao Ni, and Jiong Xia. 2024. BrainRAM: Cross-Modality Retrieval-Augmented Image Reconstruction from Human Brain Activity. In *Proceedings of the 32nd ACM International Conference on Multimedia (MM '24), October 28-November 1, 2024, Melbourne, VIC, Australia.* ACM, New York, NY, USA, 10 pages. https://doi.org/10.1145/3664647.3681296

## 1 INTRODUCTION

*"What you see is what you get."* The cognitive function of humans is closely intertwined with the visual system [51]. People capture information about the external world through their eyes and obtain corresponding perceptions through prior knowledge and memory within their minds [4, 12]. Understanding the intricate visual process occurring in the human brain has long been a hot topic in cognitive neuroscience. Within this overarching topic, decoding visual stimuli from functional magnetic resonance imaging (fMRI) represents one of the most challenging issues that has garnered extensive attention [10, 12, 20, 29, 30, 36, 44].

Early attempts [18, 29, 32] demonstrated the feasibility of decoding visual images. However, the results were primitive and abstract due to the limited expressive ability of traditional regression methods. Over the last few years, with the prosperity of Artificial Intelligence Generated Content (AIGC), a series of methods [3, 9, 20, 43, 44] adopt generative models to decode visual activities. Shen et al. [44] and Ozcelik et al. [34] have employed Generative Adversarial Network (GAN) [16] to reconstruct stimulus. More recent works [8, 23, 28, 42, 47, 49, 55] leverage diffusion models [11, 19, 39, 56] to generate high-quality images corresponding to the original visual stimuli. However, fMRI is a transient response to human brain activity, which does not adequately reflect the role of past human knowledge and memory in the cognitive process. It also inherently contains some physiological noise unrelated to visual stimuli. As shown in Fig 2, relying solely on fMRI for decoding will inevitably lead to semantic and structural inconsistency.

The recent study by Scotti et al. [42] observed that images based on the retrieval of brain activities tend to be more realistic than the reconstructed ones. However, this observation has not been thoroughly explored and leveraged. Meanwhile, Retrieval-Augmented Generation (RAG) has achieved impressive success in generative models [5, 45, 46, 52, 54, 57]. It introduces an information retrieval

**(a) Previous Methods**

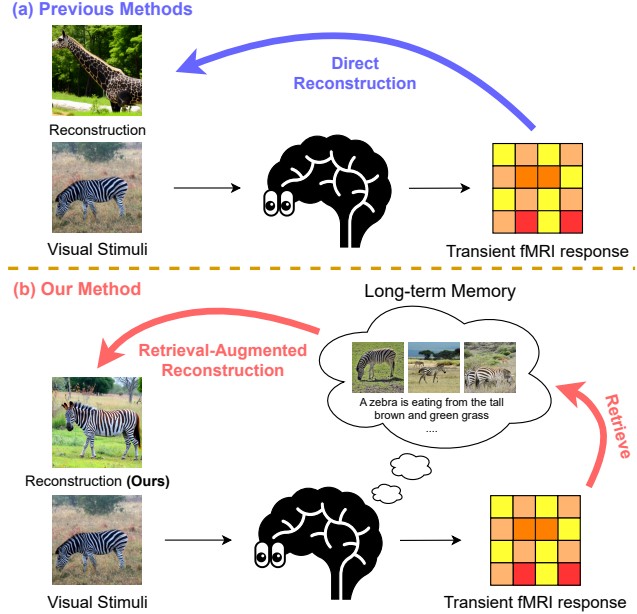

Figure 1: Different perspectives on visual stimuli reconstruction. Previous methods overlooked the transient characteristics of fMRI and did not consider the impact of human memory on visual perception.

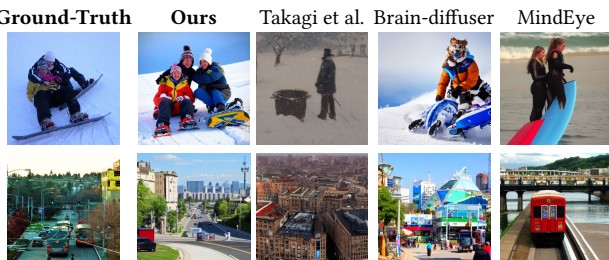

Figure 2: A brief comparison of reconstruction results from BrainRAM, Takagi et al. [49], Brain-diffuser [35], and Mind-Eye [42]. Inconsistency still exists in many recent works.

- We propose BrainRAM, a novel framework for visual stimuli reconstruction, which utilize Retrieval-Augmented Generation (RAG) and general visual-linguistic data to produce photo-realistic results.
- BrainRAM features an image-text dual-guided framework and a Retrieval-Augmentation Module (RAM). The dual-guided framework leverages both image and text features simultaneously to accurately estimate the semantic and structural aspects of the visual stimuli. The proposed RAM provides an unprecedented perspective on visual stimuli reconstruction by mimicking human memory and using retrieved samples to refine reconstruction results.
- We conduct thorough experiments to demonstrate the effectiveness of the visual-linguistic dual-guidance and retrieval-augmentation. It achieves state-of-the-art in reconstruction and retrieval compared to current reconstruction-only methods.

## 2 RELATED WORKS

### 2.1 Visual stimuli reconstruction

Neural decoding from visual stimuli has captivated researchers many years. Several early studies have demonstrated the possibility of decoding visual information from brain activity patterns measured using functional magnetic resonance imaging (fMRI). Naselaris et al. [31] showed that natural images can be reconstructed from fMRI data using a linear decoder. Kay et al. [21] used a support vector machine to decode the orientation of gratings based on activity patterns in the early visual cortex. Nishimoto et al. [32] used natural scences as visual stimuli. However, traditional visual stimuli reconstruction methods [10, 14, 21, 30–32] only rely on linear regression models to fit fMRI data with paired images. The results were abstract and blurry due to the limited expressive capabilities of regression models. In recent years, a series of studies [3, 20, 26, 43, 44] have delved into the potential of harnessing deep learning algorithms to unravel visual information within the human brain. Beliy et al. and Gaziv et al. [3, 15] employed encoder-decoder structure to align image representations with fMRI voxels. Shen et al. [44] fed the predicted features into GAN to reconstruct original visual stimuli. Ozcelik et al. [34] used a regression model to extract fMRI features and fine-tuned a pre-trained conditional big-GAN. The introduction of diffusion model [11, 19] and Contrastive

process that enhances generated results by retrieving relevant objects from a large dataset, thereby improving accuracy and robustness. This parallels human cognitive and memory functions, where individuals with rich knowledge and experience tend to describe objects more accurately than those with limited experience. Moreover, the introduction of RAG can compensate for limited training samples with the support of massive data. Therefore, we believe that RAG is an effective means to improve the quality of reconstruction.

In this paper, we propose a novel framework for visual stimuli reconstruction called BrainRAM, which aims to produce more faithful reconstructions by imitating human memory mechanisms. In stage I, we project fMRI voxels into the representation space of images and text separately, transforming them into corresponding features using diffusion priors. These features serve as preliminary estimates of visual stimuli. To further enhance alignment and alleviate inconsistency, we introduce the Retrieval-Augmented Module (RAM) to select and integrate features between preliminary estimates and retrieved samples. In stage II, we retrieve samples from the LAION-2B-en dataset [41] based on prior output features. These features, along with the retrieved samples, collaboratively predict the image and text features of the visual stimuli in RAM. We illustrate the differences between BrainRAM and previous methods briefly in Figure 1. Extensive experiments have demonstrated that our method outperforms state-of-the-art methods. Two examples in Figure 2 also showcase the superiority of BrainRAM.

In summary, our main contributions are formulated as follows:

Language-Image Pre-training (CLIP) [37] provides powerful tools for this task. Takagi et al. [49] use regression model to mapping fMRI to image and text features of Stable Diffusion [39]. Chen et al. [8] use Masked Aucoencoder (MAE) to extract fMRI feature and fine-tuned a LDM to reconstruct stimulus. Lu et al. [28] controlled the semantic and structure of the results by mapping fMRI voxels to CLIP features and VQ-VAE latent features. Scotti et al. [42] leverage diffusion prior from DALL-E 2 [38] to transform fMRI to image representations. These methods have achieved impressive results in visual stimuli reconstruction. But the above methods have limitations in image fidelity and semantic accuracy, resulting in unreliable reconstruction results.

## 2.2 Retrieval-Augmented Generation

Recently, the use of external memory to enhance large models has attracted attention. Typically, adding more training data can improve model's performance. But it is inefficient to collecting data and retraining a model with a large number of parameters. In natural language processing (NLP), Wu et al. [54] proposed a memory transformer to store information from past inputs. Querying in the memory component could improve performance many downstream tasks. Several studies [46, 52] also introduced retrieval-augmented generation in computer vision. They use the retrieved data to generate high fidelity and faithful images. RetrieveGAN [52] used a differentiable retrieval module to generate images based on scene descriptions. IC-GAN [7] trained a GAN using the neighborhood of the training image and generate samples by adjusting individual instances from the training data. KNN-diffusion [45] trained diffusion based models using large-scale retrieval methods without any textual data. The model is conditioned on text or image feature extracted by CLIP, and use k retrieved embeddings from a large dataset to generate images. RDM [5] combines small diffusion or autoregressive models with large external image datasets to form a semi parametric model. The model retrieves a set of nearest neighbors for each training image from the dataset and adjusts the diffusion model based on their CLIP embeddings.

These studies have proved that retrieving samples from large datasets can enhance the generation quality. But prior works of neural decoding only focus on aligning fMRI with image or text features directly. As far as we know, BrainRAM is the first method to integrate retrieval-augmentation into visual stimuli reconstruction, which utilize retrieved samples to achieve higher fidelity and semantic accuracy in reconstructions.

## 3 METHOD

### 3.1 Overview

In this subsection, we give a detailed analysis of the fMRI data and visual-linguistic representations that inspire our method's design.

As a non-invasive measurement, fMRI measures the blood-oxygen-level-dependent (BOLD) level in every brain voxels, indirectly reflecting the intensity of activity in brain regions. The neural activity in the brain often exhibits certain clustering characteristics in space, where similar neurons participate in processing similar information. Therefore, when a brain region is activated, it usually leads to changes in the blood oxygen levels of that region and its surrounding areas, resulting in signals with similar amplitudes between adjacent voxels in fMRI. Therefore, the spatial representation of fMRI is sparse. Besides, it is inherently noisy because fMRI not only records brain activity induced by visual stimuli, but also signals from other physiological and cognitive processes. Therefore, we need to compress fMRI voxels into a denser representation space and utilize RAG on general visual-linguistic data to overcome the inherent noise of fMRI.

Driven by the analysis above, we propose our two-stage dual-guided framework, which illustrated in Fig 3. In stage I, the fMRI signals are projected and transformed into image/text representations with diffusion prior. The predicted embedding will serve as a query to find k-nearest-neighbour of image-text pairs in LAION-2B-en dataset in stage II. Then, the retrieved data and the predicted embedding will be fed into the retrieval-augmentation module to generate a further refined embedding. Finally, the refined image/text embeddings will jointly guide the image generation in Versatile Diffusion dual-guided pipeline.

### 3.2 Stage I: Brain-visual-linguistic Consistency

As mentioned in 3.1, fMRI signals have sparse representations just like images. Therefore, it is necessary to project it onto a tighter, shared feature space through contrastive learning. The success of Contrastive Language-Image Pre-Training (CLIP) [37] proves that images and text could essentially share same representations. However, the final representation space of CLIP itself is too small ($1 \times 768$), making it difficult to recover the original details from shared features. Aligning tightly on a larger representation space will be more feasible for our goal. Meanwhile, previous studies have proved the effectiveness of diffusion prior to transfer features and reconstruct between modalities [13, 38, 42]. As a result, we adopt diffusion prior as backbone to maintain the consistency between fMRI and image/text representations.

Consider the triplet dataset $\Omega = \{S_i, V_i, T_i\}_{i=1}^{n}$ of { fMRI, image, captions }. The fMRI data $S_i \in \mathbb{R}^{1 \times N}$ is pre-processed fMRI beta values. The beta value is obtained by weighted average on the blood oxygen level response throughout the entire experiment, avoiding processing more sparse and lengthy temporal data in subsequent analysis. The fMRI data was flatten as a 1D series based on the region of interest (ROI) on visual cortex, $N$ denotes the number of voxels in the ROI. $V_i \in \mathbb{R}^{H \times W \times 3}$ denotes visual stimuli presented to the subject, and $T_i$ is corresponding COCO captions of the visual stimuli.

We adopt the pre-trained image encoder $\mathcal{E}_{img}(\cdot)$ and text encoder $\mathcal{E}_{txt}(\cdot)$ of CLIP to extract image and text features. However, instead of direct using the final outputs of the encoder ($1 \times 768$), we use the last hidden layers as the mapping target of fMRI ($257 \times 768$ for image feature, and $77 \times 768$ for text feature), avoiding overly tight spatial constraints on expression ability.

At first, the fMRI data will pass a MLP projection layer and subsequent mamba blocks $\Phi_{proj}(\cdot)$ to projected into a intermediate space (e.g. $257 \times 768$ or $77 \times 768$). Then the diffusion prior will generate corresponding target feature ($\mathcal{E}_{img}(V_i)$ or $\mathcal{E}_{txt}(T_i)$) from Gaussian noise conditioned on $\Phi_{proj}(S_i)$. In forward diffusion process, the sample at each time point is defined as:

$$\mathbf{x}_t = \sqrt{\alpha_t}\mathbf{x}_0 + \sqrt{1 - \alpha_t}\epsilon_t, \tag{1}$$

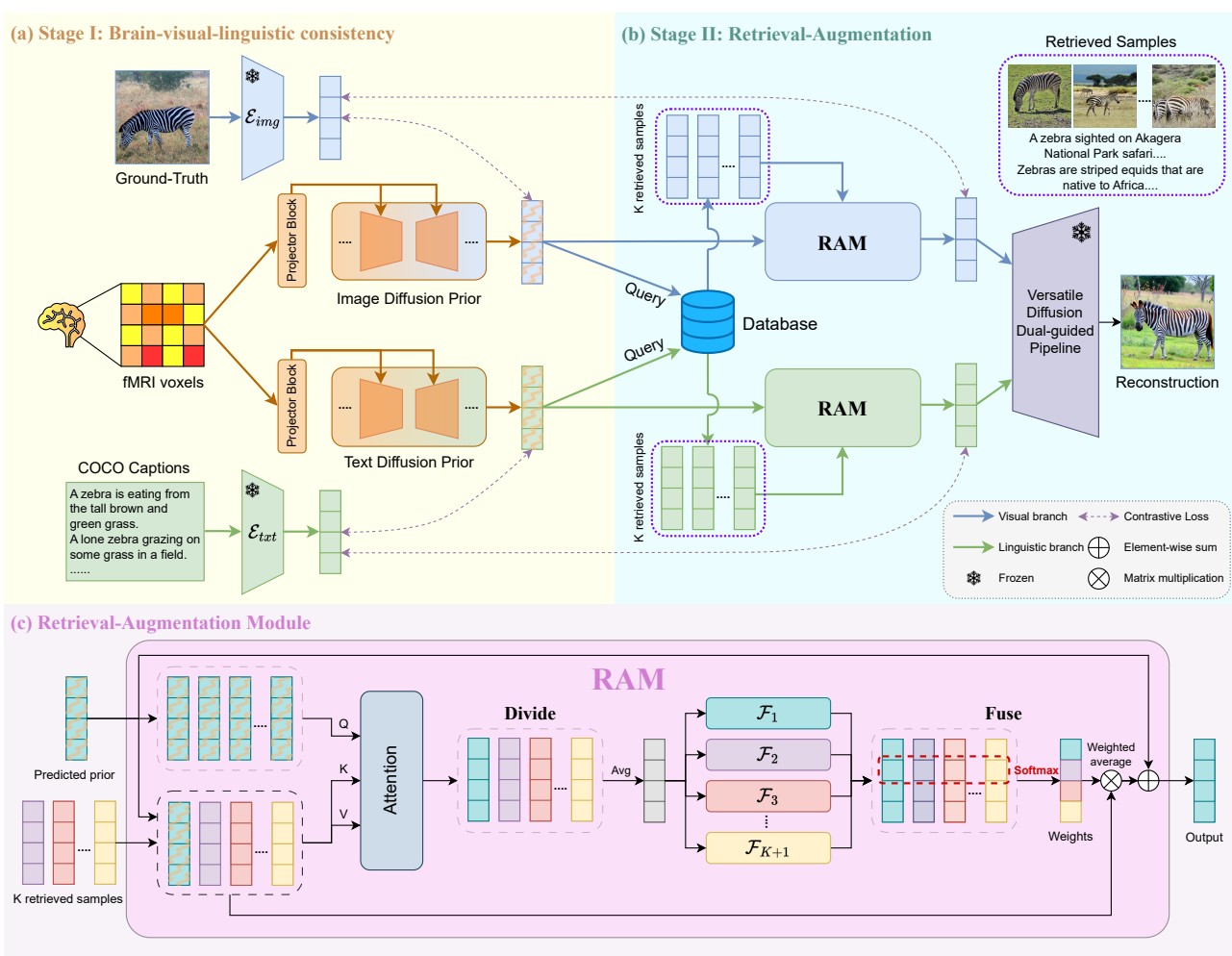

**Figure 3: Overview of our proposed BrainRAM.** $\mathcal{E}_{img}$ and $\mathcal{E}_{txt}$ **represents CLIP image encoder and text encoder, respectively. In stage I, we transform fMRI voxels into corresponding image and text features via diffusion prior. In stage II, we use RAM to combine prior features and retrieved samples, and then refine the final output.**

where $\mathbf{x}_0$ is target GT feature, $\alpha_t$ represents the noise variance schedule and $t \in \{0, 1, \dots, T\}$ [39]. The inverse diffusion process will apply a attention-based U-Net $f_\theta(\mathbf{x}_t, t, \Phi_{proj}(S_i))$ to gradually recover the original input $\mathbf{x}_0$ from noisy $\mathbf{x}_T$. Specifically, the conditional information $\Phi_{proj}(S_i)$ is incorporated in the cross-attention module of $f_\theta$:

$$\begin{aligned} \text{CrossAttention}\,(Q, K, V) &= \text{softmax}\left(\frac{QK^T}{\sqrt{d}}\right)V, \\ Q = W_Q^{(i)} \varphi_i\,(\mathbf{x}_t)\,, K &= W_K^{(i)} \Phi_{proj}(S), V = W_V^{(i)} \Phi_{proj}(S), \end{aligned} \tag{2}$$

where $\varphi_i\,(\mathbf{x}_t)$ indicates intermediate features of U-Net [40] and $W_Q^{(i)}, W_K^{(i)}, W_V^{(i)}$ denote trainable attention projection matrices.

We use InfoNCE loss [33] to align the representations of fMRI to target features:

$$\mathcal{L}_{\text{InfoNCE}} = -\frac{1}{B} \sum_{i=1}^{B} \log \frac{\exp\,(p_i \cdot c_i / \tau)}{\sum_{j=1}^{B} \exp\,(p_i \cdot c_j / \tau)}, \tag{3}$$

where $p$ denotes the output of diffusion prior, $c$ denotes aligned target, $\tau$ is a temperature hyperparameter. It has 1 positive and $B - 1$ negative samples in batch size of $B$. We also incorporate Mean Square Error (MSE) loss to improve retrieval performance in subsequent stage, with a hyperparameter $\gamma$ to balance these two losses in Eq 6.

Due to the limitation of limited number of training samples of brain-image-text triplet compared to the contrastive learning in CLIP, we adopt the MixCo [22] augmentation strategy to improve the robustness. Given two fMRI data $S_i$ and $S_k$, the convex mixture

are given by:

$$S_{mix_{i,k}} = \lambda_i \cdot S_i + (1 - \lambda_i) \cdot S_k, \qquad (4)$$

where $S_k$ is a arbitrary sample shared the same batch with $S_i$. Let $p_i^*$ be the diffusion prior output from $S_{mix_{i,k}}$, and $c$ is the corresponding target feature, the MixCo loss is given by:

$$
\begin{aligned}
\mathcal{L}_{\text{MixCo}} = -\sum_{i=1}^{n} \Bigg[ & \lambda_i \cdot \log \frac{\exp\left(p_i^* \cdot c_i / \tau\right)}{\sum_{j=0}^{K} \exp\left(p_i^* \cdot c_j / \tau\right)} \\
& + (1 - \lambda_i) \cdot \log \frac{\exp\left(p_i^* \cdot c_k / \tau\right)}{\sum_{j=0}^{K} \exp\left(p_i^* \cdot c_j / \tau\right)} \Bigg].
\end{aligned}
\qquad (5)
$$

With introduced hyperparameter $\beta$ to control the intensity of $\mathcal{L}_{\text{MixCo}}$, the total loss of stage I can formulated as:

$$\mathcal{L}_{\text{total1}} = \gamma \mathcal{L}_{\text{InfoNCE}} + (1 - \gamma)\mathcal{L}_{\text{MSE}} + \beta \mathcal{L}_{\text{MixCo}}. \qquad (6)$$

## 3.3 Stage II: Retrieval-Augmentation

In this stage, we proposed Retrieval-Augmentation Module (RAM), which illustrated in Fig. 3 (c), to incorporate massive real-world data into the generated results. Unlike other retrieval-augmented image generative models, which require to train the entire diffusion model, we utilize RAM in embedding level that can directly applied to the existing model. To enable the module adaptively select feature from the retrieved data, we perform an automatic selection contain two operations: *Divide* and *Fuse* [25]. Note that the diffusion prior and fMRI projection block are frozen in this stage.

**Divide.** Let $p \in \mathbb{R}^{T \times D}$ be the output of diffusion prior, where $T$ is the number of tokens (257 for image and 77 for text) and $D$ is the dimension of tokens (typically 768). The $p$ will serve as a query in LAION-2B-en, and retrieve $K$ embeddings $\{c_i\}_{i=1}^{K}$ based on the similarity between $c_i$ and $p$. Every $c_i$ is performed cross-attention with $p$ separately, with query from $p$, key and value from $c_i$. Note that $p$ also performed self-attention too. These operations will generate $k+1$ attention maps, denoted as:

$$
\begin{aligned}
u_{c_i} &= \text{CrossAttention}\,(Q_p, K_{c_i}, V_{c_i}), \\
u_p &= \text{SelfAttention}\,(Q_p, K_p, V_p), \\
\mathbf{U} &= \left[u_p, u_{c_1}, u_{c_2}, \dots, u_{c_K}\right]^T,
\end{aligned}
\qquad (7)
$$

where $u_p, u_{c_i} \in \mathbb{R}^{T \times D}$, $\mathbf{U} \in \mathbb{R}^{(K+1) \times T \times D}$. The weights are shared between cross-attention and self-attention.

**Fuse.** To integrate information from $K+1$ features, we averaged them firstly:

$$\bar{u} = \frac{1}{K+1}\left(u_p + \sum_{i}^{K} u_{c_i}\right). \qquad (8)$$

Then, we apply different tiny MLP $\mathcal{F}_i(\cdot)$ to extract sample-specific information from the compact $\bar{u}$ separately:

$$
\mathbf{Z} = \begin{bmatrix} \mathcal{F}_1(\bar{u}) \\ \mathcal{F}_2(\bar{u}) \\ \dots \\ \mathcal{F}_{K+1}(\bar{u}) \end{bmatrix}, \quad i = 1, 2, \dots, K+1. \qquad (9)
$$

where $\mathbf{Z} \in \mathbb{R}^{(K+1) \times T}$.

To acquire the guidance for adaptive selection from different samples, we apply softmax to the first dimension of $\mathbf{Z}$:

$$\tilde{\mathbf{Z}} = \text{softmax}(\mathbf{Z}). \qquad (10)$$

where the first dimension of $\tilde{\mathbf{Z}}$ are weights for feature from different tokens. Let $q \in \mathbb{R}^{T \times D}$ be the final output of RAM, every element in $q$ is given by:

$$q_{ij} = \sum_{k}^{K+1} \sum_{i}^{T} \tilde{\mathbf{Z}}_{ki} \cdot \mathbf{U}_{kij}, \quad j = 1, 2, \dots, D, \qquad (11)$$

which represents weighted average for every token on different $u$.

To ensure the reconstruction results won't get worse with RAM, we consider the optimization objective of $q$ as residual between prior output $p$ and ground truth embedding $c$, which means $c = p+q$ in the most ideal situation. Besides, the initial weight of the last dense linear layer in Attention is zero-initialized, which means the initial output of RAM is zero, to protect $p$ from the damage of random initialization. Considering that the residual on embedding is relatively small, we use the loss with smaller $\gamma$ value compared to stage I:

$$\mathcal{L}_{\text{total2}} = \gamma \mathcal{L}_{\text{InfoNCE}} + (1 - \gamma)\mathcal{L}_{\text{MSE}}. \qquad (12)$$

## 4 EXPERIMENTS

### 4.1 Dataset

We employ the Natural Scenes Dataset (NSD) [1] for this study. NSD is the largest known dataset that using 7T scanner to acquire visual-evoked fMRI response from 8 subjects. Each subject was presented about 10,000 different natural images multiple times during 30 to 40 sessions, with whole-brain gradient-echo EPI at 1.8mm isotropic resolution and 1.6s TR for scanning. The natural images in the paradigm are sourced from Common Objects in Context dataset (COCO) [27], which have 5 or 6 captions correspondingly for each image. In this study, we only focused on 4 subjects (subj01, subj02, subj05, and subj07) out of 8, since they have completed all scanning sessions. Same as other experiment, we use *nsdgeneral* mask to select ROI, which provide a general ROI that covering voxels responsive to the NSD experiment in the posterior aspect of cortex.

### 4.2 Implementation Details

We follow the same train-test splits in NSD, with 8,859 image stimuli for training and 982 for testing for each subject. Also, we averaged response for image with multiple fMRI trials, and the target text feature are averaged on multiple captions.

In stage I, we training image diffusion prior and text diffusion prior separately for 180 epochs with batch size of 32. We use AdamW optimizer with default parameters and one-cycle learning rate $2.5 \times 10^{-4}$. The loss hyperparameter $\gamma$ and $\beta$ are set as 0.8, 0.3 respectively. In stage II, we use clip-retrieval tool [2] to retrieve data in LAION-2B-en. Due to the randomness of diffusion prior output, we reconstruct 4 priors per image stimuli, and each prior was retrieved for another 4 samples. We apply zero-initialization to the last dense linear layer of Attention in RAM. Training for 120 epochs with batch size of 256, and default AdamW optimizer with fixed learning rate $2.5 \times 10^{-4}$. The loss hyperparameter $\gamma$ is 0.2.

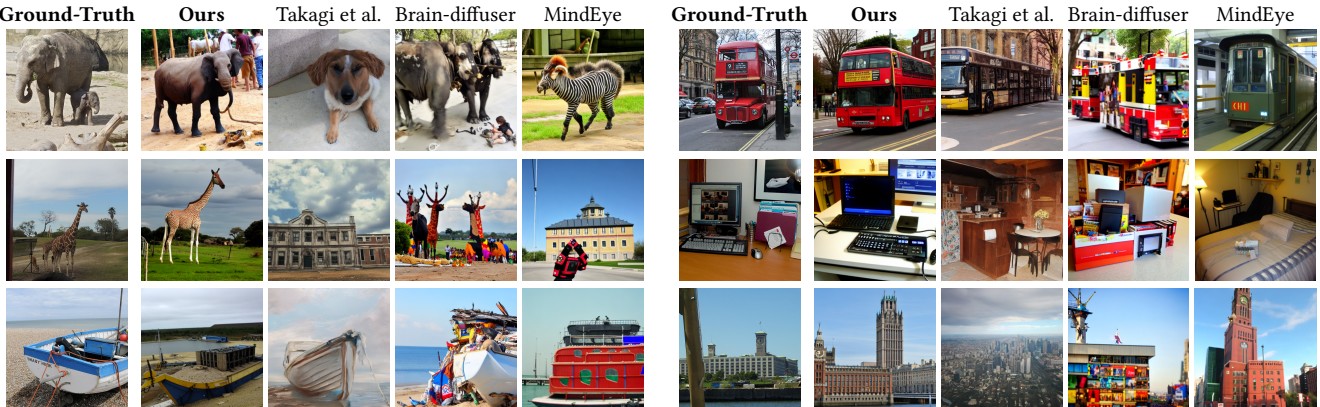

Figure 4: Visual comparison of reconstruction results.

| Method | Low-Level | | | | High-Level | | | |
|---|---|---|---|---|---|---|---|---|
| | PixCorr↑ | SSIM↑ | Alex(2)↑ | Alex(5)↑ | Incep↑ | CLIP↑ | Eff↓ | SwAV↓ |
| Lin et al. [26] | - | - | - | - | 78.2% | - | - | - |
| Takagi et al. [49] | .148 | .285 | 71.9% | 74.8% | 64.5% | 63.3% | .953 | .684 |
| Gu et al. [17] | .150 | .325 | - | - | - | - | .862 | .465 |
| Brain-diffuser [35] | .136 | .321 | 81.1% | 85.9% | 75.9% | 73.8% | .895 | .548 |
| MindEye [42] | **.182** | **.358** | 88.9% | 92.4% | 88.2% | 90.0% | .719 | .422 |
| **BrainRAM (Ours)** | .176 | .342 | **89.9%** | **95.7%** | **92.6%** | **94.1%** | **.666** | **.381** |

Table 1: Quantitative evaluation of reconstruction results. The best performance is highlighted in bold, while the second performance is highlighted with underline. Missing values are from papers not reporting all metrics or metrics being non-applicable.

## 4.3 Evaluation Metrics

Due to inherent noise in current measurement method, it is impossible to reconstruct all details of the image from fMRI with complete accuracy. As a result, we mainly focus on semantic consistency in this task. But for the sake of fairness, we still use some low-level metrics in accordance with previous studies [35, 42]. Specifically, for high-level metrics, we use two-way comparison of the last hidden layer of CLIP image encoder $\mathcal{E}_{img}(\cdot)$, denoted as CLIP [37]. EffNet-B and SwAV indicate the average correlation distance gathered from EfficientNet-B1 [50] and SwAV-ResNet50 [6], correspondingly. Inception represents the two-way comparison of the last pooling layer of InceptionV3 [48]. In low-level metrics, the Structural Similarity Index (SSIM) [53] measures the similarity between two images. PixCorr indicates pixel-level correlation between the reconstructed and ground-truth images. AlexNet(2) and AlexNet(5) represent two-way comparisons of the second and fifth layers of AlexNet [24], respectively.

## 5 RESULTS

## 5.1 Reconstruction Results

In this section, we compared our BrainRAM with other five state-of-the-art methods: Lin et al. [26], Takagi et al. [49], Gu et al. [17], Brain-diffuser [35], and MindEye [42]. Due to some methods' results

were based on experiments conducted on subj01, for the sake of fairness, if not specified, all our experiments were conducted on subj01. The quantitative performance of each model are presented in Tab 1. Our method outperforms in all high-level metrics and part of low-level metrics. Especially in CLIP similarity, BrainRAM earn 4.1%, 20.3%, and 32.8% performance gains over previous state-of-the-art MindEye [42], Takagi et al. [49], and Brain-diffuser [35], which proves that our method has achieved significant success in high-precision semantic information extraction.

In addition to quantitative metrics, several visual cases are also illustrated in Fig 4 that are consistent with numerical findings. For an example, in the first row of Fig 4, other methods did not accurately capture semantic information, which misidentified the elephant in GT as a dog (Takagi et al. [49]) or zebra (MindEye [42]). Alternatively, Brain-diffuser [35] can't accurately reconstruct the structure of the elephant. Besides, for the Red double-decker bus, other methods can only reconstruct it as a public transportation vehicle. In contrast, BrainRAM precisely reconstructs the red double-decker bus with its unique structure and semantic information. These results indicate that BrainRAM could produce results that are consistent with the visual stimuli in terms of semantics and structure.

| Ground-Truth | Ours | | | | MindEye | | | |
|---|---|---|---|---|---|---|---|---|
| | Top 1 | Top 2 | Top 3 | Top 4 | Top 1 | Top 2 | Top 3 | Top 4 |

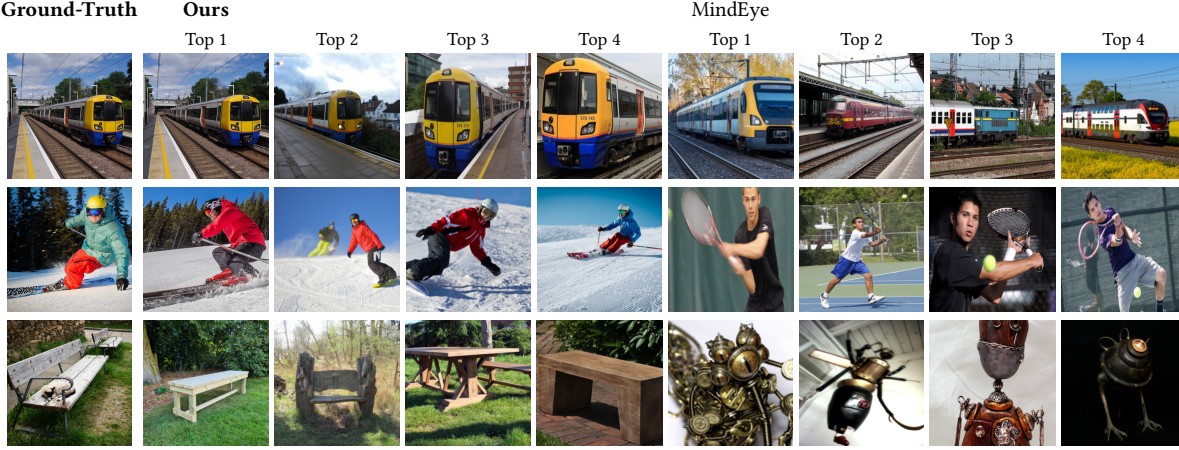

**Figure 5: Visual comparison of retrieved results on LAION-2B-en. The first column represents the original visual stimuli presented to the subject. The second to fifth columns and sixth to ninth columns represents retrieved images based on MindEye and BrainRAM, respectively. The similarity of the four retrieved images is arranged in descending order from left to right.**

| Method | High-Level | | | | Image↑ | Brain↑ |
|---|---|---|---|---|---|---|
| | Incep↑ | CLIP↑ | Eff↓ | SwAV↓ | | |
| Lin et al. [26] | - | - | - | - | 11.0% | 49.0% |
| Brain-diffuser [35] | - | - | - | - | 21.1% | 30.3% |
| MindEye [42] | 86.7% | 86.5% | .724 | .457 | 89.8% | 85.3% |
| Ours (w/o RAM) | 87.4% | 88.9% | .737 | .481 | 90.1% | 84.6% |
| **BrainRAM (Ours)** | **88.1**% | **91.0**% | **.726** | **.444** | **93.4**% | **90.3**% |

**Table 2: Quantitative evaluation of retrieval results. The first four columns refer to high-level metrics computed on retrieved images from LAION-2B-en. The last two columns represent image retrieval and fMRI retrieval performance on the test set.**

## 5.2 Retrieval Results

In this subsection, we will demonstrate the performance of test set retrieval and LAION retrieval separately.

**Test set retrieval.** Test set retrieval contain fMRI retrieval and image retrieval. Image retrieval means retrieving the image embedding with the highest CLIP similarity based on predicted output embedding from fMRI in the test set. The image retrieval is considered correct if a paired image embedding is retrieved, and vice-versa for fMRI retrieval. Follow the same settings of previous studies [26, 42], we randomly select 300 image-fMRI pairs in the test set of NSD, and computing image embeddings and corresponding output embeddings from fMRI voxels. The quantitative results are presented in Tab 2. Compared to previous studies, our retrieval accuracy improved significantly for both image and brain retrieval ways, with 3.6% and 5.7% superiority over MindEye [42].

**LAION retrieval.** LAION retrieval refers to image retrieval on LAION-2B-en [41], which can reflect the model's potential to capture visual information from fMRI from another perspective. For an output embedding, we could query K-nearest neighbours to retrieve images most relevant to the model's prediction. Given the emphasis on semantic relevance in embedding-based retrieval, we only focus high-level image evaluation metrics. We calculate 16 output embeddings per fMRI data, and retrieve 16 images with highest CLIP similarity in correspondence to the outputs. Experiment results in Tab 5 shows that our BrainRAM outperforms MindEye [42] in most of high-level metrics. And indicate introducing text guidance is effective in improving the capability of capture semantic information, compared to MindEye solely relied on image guidance. Also, most of performance still remains better than MindEye even without our proposed RAM. The visual comparison of retrieved results in Fig 5 also demonstrates the effectiveness of our method.

## 5.3 Ablation Studies

In this subsection, we further conduct comprehensive experiments to verify the validity of each component in the model. All quantitative performance can be found in Tab 3.

**Effect of model architecture.** Obviously, as shown in the Tab 3, the introduction of RAM significantly increases the reconstruction performance of the model. It can also be observed that even using predicted features directly to reconstruct visual stimuli in stage I can surpass MindEye in most of high-level metrics. We also conduct experiments to compare our proposed RAM with other feature fusion methods: Average, 1x1 Convolution, and Cross-Attention. Average means directly average on all features. 1x1 Convolution represents use a 1x1 convolution kernel to merge all features. Cross-Attention means the prior output serves as query, and other retrieved embeddings serve as keys and values. Presented in Tab 4, RAM surpasses all other feature fusion methods in all metrics. RAM leverages *Divide* and *Fuse* operations to select features from retrieved embeddings and prior output, thus enhance reconstruction using retrieved information. It also ensures that the performance of the model will not decrease after adding RAM.

As mentioned in Sec 3.2, we used several Mamba blocks after the MLP projection layer, instead of purely using MLP in the fMRI

| Variants | | | | | Low-Level | | | | High-Level | | | |
|---|---|---|---|---|---|---|---|---|---|---|---|---|
| MambaBlock | RAM | ImageGuide | TextGuide | Dataset | PixCorr↑ | SSIM↑ | Alex(2)↑ | Alex(5)↑ | Incep↑ | CLIP↑ | Eff↓ | SwAV↓ |
| | | ✓ | ✓ | N/A | .161 | .317 | 83.2% | 89.3% | 85.1% | 90.1% | .823 | .412 |
| ✓ | | ✓ | ✓ | N/A | **.183** | .328 | 84.3% | 93.4% | 88.3% | 91.2% | .711 | .412 |
| ✓ | ✓ | ✓ | | 2B-en | .170 | .337 | 87.6% | 92.7% | 85.1% | 90.1% | .823 | .419 |
| ✓ | ✓ | | ✓ | 2B-en | .165 | .320 | 82.9% | 88.4% | 90.1% | 91.3% | .709 | .397 |
| ✓ | ✓ | ✓ | ✓ | 400M | .174 | **.345** | 89.3% | 93.2% | 90.3% | 92.2% | .679 | .393 |
| ✓ | ✓ | ✓ | ✓ | 2B-en | .176 | .342 | **89.9%** | **95.7%** | **92.6%** | **94.1%** | **.666** | **.381** |

Table 3: Ablation study on model architecture, dual-guidance, and retrieval dataset. The column of Dataset refers to the retrieval LAION dataset the model based on, N/A means retrieval didn't involve in this process.

| Fusion method | Low-Level | | | | High-Level | | | |
|---|---|---|---|---|---|---|---|---|
| | PixCorr↑ | SSIM↑ | Alex(2)↑ | Alex(5)↑ | Incep↑ | CLIP↑ | Eff↓ | SwAV↓ |
| Average | .153 | .298 | 82.3% | 87.4% | 84.3% | 85.2% | .806 | .526 |
| 1x1Conv | .162 | .326 | 86.5% | 92.8% | 90.8% | 89.7% | .687 | .407 |
| Cross-Attention | **.178** | .332 | 87.9% | 93.9% | 91.4% | 90.2% | .677 | .392 |
| RAM (Ours) | .176 | **.342** | **89.9%** | **95.7%** | **92.6%** | **94.1%** | **.666** | **.381** |

Table 4: Effectiveness of RAM. We conduct experiments using different feature fusion methods to integrate prior outputs and retrieved samples.

projector. Experiments in the first and second row of Tab 3 have shown that replace MLP backbone with Mamba blocks achieved improvement in high-level metrics.

**Effect of image text dual-guidance.** Intuitively, text contains richer semantic information, thus could enhance the semantic fidelity of reconstructed image. Results from the third and fourth row in Tab 3 also verified that idea. Our BrainRAM maintain higher semantic superiority than solely relied on image guidance. It is also impractical to reconstruct image with text guidance only. Since text is on semantic-level, which may not be able to provide details in the image. Furthermore, due to the ambiguity of language representation, RAM has limited effectiveness in this situation.

**Effect of retrieval dataset.** When using RAM, we default to retrieve sample on LAION-2B-en. To examine the impact of retrieval datasets on model performance, we conduct experiments with different retrieval dataset. Specifically, we re-trained RAM on LAION-400M, and results could be found in the second to the last row of Tab 3. We observed that when retrieving on LAION-400M, the model's performance also increased, but not significant like retrieving on LAION-2B-en. On the one hand, smaller data size reduces data diversity in the retrieved samples. If retrieving on a larger dataset, it may even be possible to retrieve ground-truth stimuli based on prior output (like in Fig 5), which would be very helpful for reconstruction. On the other hand, captions from LAION-2B-en are only in English. LAION-400M contains multi-language captions that may cause confusion for CLIP text encoders, since CLIP are trained on image-text pairs with English captions.

## 6 CONCLUSION

In this study, we propose a two-stage dual-guided visual stimuli reconstruction framework called BrainRAM. In stage I, we first project the fMRI signal and then convert it into CLIP text image features through diffusion prior. In stage II, we retrieve samples from LAION-2B-en to refine prior outputs, and use Versatile Diffusion to reconstruct visual stimuli. Experiments have shown that BrainRAM outperforms current state-of-the-art methods both qualitatively and quantitatively in reconstruction and retrieval. In addition, we provide a completely new perspective for this task: introducing retrieval-augmented generation into visual stimuli reconstruction can help improve image fidelity and semantic accuracy. For future work, we believe that cross-subject decoding would be a promising direction. By utilizing the responses of different subjects in multiple datasets, we can further understand the visual mechanisms of entire humanity.

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
