# OpenReview forum: "BrainRAM: Cross-Modality Retrieval-Augmented Image Reconstruction from Human Brain Activity"
_acmmm.org/ACMMM/2024/Conference — MM2024 Oral_

### Official Review · Reviewer_4nZu · 2024-05-07

**Rating:** 4
**Confidence:** 4

**Summary:**

The manuscript introduces BrainRAM, an innovative dual-stage framework for visual stimulus reconstruction from brain activity. Drawing inspiration from memory mechanisms, the study employs a Retrieval-Augmented Module (RAM) and diffusion priors to enhance the quality of reconstructed images. The methodology involves an initial mapping of fMRI voxels into a latent space for semantic and structural estimation, followed by data retrieval from the LAION-2B-en dataset and RAM-based refinement, achieving superior reconstruction quality.
The study demonstrates that BrainRAM outperforms current state-of-the-art methods both qualitatively and quantitatively, providing a novel perspective for research in visual reconstruction. However, challenges such as the scarcity of large-scale datasets and the noise inherent in non-invasive measurement techniques are acknowledged and warrant further investigation. Overall, the research significantly contributes to understanding the neural mechanisms of the human visual system and paves the way for future advancements in the field.

**Strengths:**

1. This paper presents a two-stage dual-guided framework for visual stimuli reconstruction, achieving state-of-the-art (SOTA) performance on the largest neuroimaging dataset NSD.

2. The experimental results demonstrate that BrainRAM is capable of reconstructing realistic natural images.

3. The framework proposed by the authors has achieved commendable results in both reconstruction and retrieval tasks.

4. The authors have conducted thorough ablation studies to validate the effectiveness of each component proposed, as well as to demonstrate that utilizing a larger database can enhance the model's performance.

**Limitations:**

1. The authors have compared their work against baselines that do not employ retrieval augmentation techniques within the paper. It is recommended that the authors also benchmark their method against the performance of the work that utilizes retrieval augmentation, as detailed in the provided reference (https://arxiv.org/pdf/2312.07705).

2. The manuscript's writing quality is subpar, particularly in the 'Contribution' section (lines 150-166), which appears to merely describe the model architecture and performance. It is challenging to discern the authors' true contributions. It is recommended that the authors consolidate these points into a single, clear statement that explicitly highlights their distinct contributions.

3. The section 3.1 'Overview' contains inaccuracies. Among all methods of recording brain responses, fMRI is considered to have a relatively high spatial resolution. However, its temporal resolution is low due to the fact that it captures the slower Blood Oxygen Level Dependent (BOLD) response. Additionally, in the second paragraph, you mention the low signal-to-noise ratio of the fMRI signal, yet the manuscript does not propose a superior denoising algorithm to address this issue. Furthermore, the transition with 'Driven by...' in the third paragraph is inappropriate and requires careful revision.

4. The authors have not made the code publicly available, which raises concerns regarding the reproducibility of the findings presented in the manuscript.

**Suitability:**

3

---

### Official Review · Reviewer_2PLG · 2024-05-21

**Rating:** 4
**Confidence:** 3

**Summary:**

This paper introduced a novel method to reconstruct images from brain activities represented by fMRI. The authors retrieve images from a large image dataset using fMRI embeddings aligned with images and texts respectively. The retrieved images will be combined with the fMRI embeddings as guidance for a versatile diffusion model. Superior performance was shown compared with the previous work.

**Strengths:**

The paper is solid as an ML paper with interesting ideas and solid experiments. The organization and writing are easy to follow.

**Limitations:**

1. The high-level semantics metric only uses a 2-way test, which has a chance level of 50%. More distinguishable metrics like a 50-way or 100-way test should also be used as a supplement.

2. The claim that the proposed RAM method considers the impact of human memory on visual perception should be careful and better elaborated. This claim may not be scientifically correct. I agree that human memory could have an impact on visual perception, but the proposed RAM method did not consider the "actual human memory", as the decoding is also based on transient fMRI signals. If I understand correctly, retrieving images from a large dataset using fMRI does not consider the human memory encoded in other areas of our brain. The proposed RAM method is more of an analogy of human memory rather than actually considering human memory.

3. Using retrieved images in the reconstruction directly could be controversial because the retrieval dataset may contain groundtruth images.   According to my knowledge, Scotti et al. [41] used the retrieval task to show that the reconstruction is not just a classification.

4. Some minor typos:
- Line 278. Should it be "embedding will be fed into.."?
- Line 458. Should it be "require to train" ?

5. Could the author elaborate more on Line 503? Why the reconstruction would get worse with RAM?

**Suitability:**

3

---

### Official Review · Reviewer_b6da · 2024-05-24

**Rating:** 5
**Confidence:** 3

**Summary:**

The paper proposes a two-stage dual-guided framework for fMRI visual decoding and image reconstruction. This method uses fMRI as conditional embeddings, leveraging the prior knowledge of diffusion models to align fMRI with both images and text separately, and refines it through retrieval enhancement. Ultimately, the image is reconstructed using versatile diffusion.

**Strengths:**

1. This method simulates the human memory mechanism, providing a new perspective for studies on the understanding and reconstruction processes of human visual cognition.

2. The method effectively utilizes diffusion priors and versatile diffusion, which are absent in previous similar works.

3. It is a novel work, and the experiments are solid.

4. The paper is well-written.

**Limitations:**

1. In Sec. 3.2, the loss hyperparameter 'α' in equation (6) overlaps somewhat with the variable in the noise variance schedule in equation (1).

2. It is recommended to supplement the implementation details with the hardware platforms used in the experiments.

3. The motivation for using versatile diffusion instead of other models seems not clear enough.

**Suitability:**

3

---

### Meta-Review · Area_Chair_oroW · 2024-07-02

**Recommendation:** Accept (Oral)
**Confidence:** 4

**Metareview:**

The rebuttal letter well address the concerns from the reviewers. All reviewers tend to accept this paper.